

# Thermography in ergonomic assessment: a study of wood processing industry workers

Denise Ransolin Soranso[1], Luciano José Minette[2], Marcio Marçal[3], João Carlos Bouzas Marins[4], Stanley Schettino[5], Roldão Carlos A. Lima[6] and Michel Oliveira[7]

[1] Institute of Production Engineering and Management, Federal University of Itajubá, Itajubá, Minas Gerais, Brazil
[2] Department of Production Engineering and Mechanics, Federal University of Viçosa, Viçosa, Minas Gerais, Brazil
[3] Department of Physiotherapy, Federal University of Vales do Jequitinhonha and Mucuri, Diamantina, Minas Gerais, Brazil
[4] Department of Physical Education, Federal University of Viçosa, Viçosa, Minas Gerais, Brazil
[5] Institute of Agrarian Sciences, Federal University of Minas Gerais, Montes Claros, Minas Gerais, Brazil
[6] School of Agriculture, São Paulo State University, Botucatu, São Paulo, Brazil
[7] Center for Agricultural Sciences and Engineering, Federal University of Espírito Santo, Jerônimo Monteiro, Espírito Santo, Brazil

Corresponding author
Roldão Carlos A. Lima,
roldao.carlos@outlook.com

## ABSTRACT

**Background**. Workers in the wood processing industry perform activities that demand great physical and ergonomic demands, which favors the emergence of inflammatory processes and in turn the occurrence of heat regions in the body, thus making it possible to assess the inflammatory level by means of temperature gradients. This study aimed to evaluate the use of thermography as an ergonomic analysis tool to identify regions with musculoskeletal overload in workers in a wood processing industry.

**Methods**. The study was conducted with nine workers in the central-west region of Brazil. The evaluations to obtain the thermographic images were carried out before the beginning of the workday, on Monday (day I) and on Friday (day II), in order to verify the overload regions in the accumulation of days worked. The thermal images were collected in an acclimatized room with controlled conditions where the participants remained with the upper part of their bodies bare for acclimatization, and then the lumbar and scapular regions were evaluated. The images were obtained using the FLUKE TI 400 Thermal Imager, with analysis using the SmartView software program to demarcate the body regions of interest.

**Results**. The mean temperature values obtained on day I did not significantly differ from the mean values obtained on day II. Qualitative analysis showed thermal patterns with high temperature at the same points on both evaluated days. Although the thermographic analysis performed in this study cannot provide definitive results, they generally helped to provide evidence for a more accurate diagnosis in the evaluated workers.

# INTRODUCTION

The evolution of pain to the chronic state, mainly in the spinal region, occurs in activities which require physical strength, being one of the main causes of temporary or permanent incapacity for workers in Brazil. Thus, it may be responsible for leading workers to resort to the social security system, requesting disability retirement (*Treede et al., 2019*; *Grant et al., 2019*; *Sündermann, Flink & Linton, 2020*).

It is known that pain symptoms arising from injury trigger inflammatory processes that can simultaneously promote the occurrence of heat regions in the body as a result of increased metabolism. Thus, it is possible to assess the inflammatory level through temperature gradients (*Brioschi et al., 2007*; *Machado et al., 2009*).

In medicine, the evaluation of physiological responses associated with skin temperature in order to identify regions of the body with lesions may be performed using thermography (*Gold, Cherniack & Buchholz, 2004*; *Santos et al., 2014*; *Chandler, 2015*; *Côrte & Hernandez, 2016*). This is a non-invasive method used to record gradients and body thermal patterns through thermographic images which help to early recognize the beginning of an inflammatory process which has not yet presented classic signs and symptoms, acting in a preventive way (*Magalhaes, Vardasca & Mendes, 2018*; *Gefen et al., 2019*; *Jimenez-Pavon et al., 2019*).

Several factors influence variations in body temperature, which may be directly associated with the metabolic process and activities of the synaptic nervous system in organs. Thus, thermal analysis makes it possible to observe the physiological changes that can occur in the human body from temperature variation (*Côrte & Hernandez, 2016*; *Tan & Knight, 2018*; *Geneva et al., 2019*).

Thermography has also been applied in studies aimed at identifying the variation in skin temperature in muscle groups of workers throughout the development of their work activities (*Oliveira et al., 2018*; *Uchôa et al., 2018*; *Moreira-Marconi et al., 2019*), standing out as a technique which can be used to evaluate ergonomic factors at work.

Wood sector activities in tropical forest regions in Brazil are commonly carried out in inappropriate thermal conditions due to the hot and humid climate, demanding high energy expenditure and physical effort from the workers. These workers often operate and handle machines and equipment that produce high levels of noise and vibration, in addition to adopting postures which can be harmful to the body given the constant lifting, handling and transport of loads weighing above the tolerable limits (*Vasconcelos et al., 2019*; *Lima et al., 2019*; *Lima et al., 2022*; *Schettino et al., 2021*).

Considering that wood processing industry activities in tropical forest regions in Brazil are highly dependent on human labor for their accomplishment, thermography can be a helpful technique to ergonomically evaluate workers subjected to conditions which favor the emergence of musculoskeletal diseases in these cases. Its study becomes interesting, as it can contribute to establish preventive actions to protect workers' health.

In this context, this study was based on the hypothesis that the accumulation of working days associated with the short recovery time between one workday and another can contribute to muscle overload in work activities, such as those performed by workers in a

**Table 1 Description of operations performed by workers in the wood processing industry.**

| Operations | Description of the operation |
| --- | --- |
| Primary split | This refers to the process of reducing whole logs through longitudinal cuts into smaller parts which can be boards, boards or rectangular or square pieces. The wood logs are moved by the log carriage to the main sawing machine, called a single vertical band saw. This operation normally involves three workers, one being responsible for operating the band saw and the other two for fixing the log in the log carrier, removing and disposing of the boards generated in the processing on a ramp that supplies the secondary split. |
| Secondary split | This corresponds to the activity performed after the primary split, with the purpose of reducing the size or defining the final dimensioning of the wooden pieces. |
| Secondary stripping | This is performed using circular saws, and aims to regularize the final length of the wooden pieces according to the standard for sale. This task is normally carried out by two workers, one being responsible for operating the circular saw and the other for removing the pieces of wood and placing them close to the site, where manual stacking/packing is later carried out by other workers. |
| Manual stacking | This consists of forming piles of sawn wood, for which the pieces of each pile have the same dimensions. Two workers manually load the pieces, holding each one at one end, and deposit them in the place where the piles are being formed. |

wood processing industry which demands great physical demands. It is possible that this work model provides inflammatory processes in body regions that are highly requested during the performance of activities, and therefore a significant increase in the body temperature of these places can occur.

In turn, it is believed that the body temperature of the most requested regions is higher at the end of the work week (Friday), when the worker has had an accumulation of days developing their activities associated with a short rest interval ($\pm$12h). This is in contrast to the beginning of the working week (Monday) when the rest period is longer ($>$48 h) because the workers do not perform work activities on Saturday or Sunday, and thereby they have more time for possible muscle recovery and consequently a reduction of inflammatory processes.

In the case of workers in the wood processing industry, the scapular and lumbar body regions tend to be the most used due to the characteristic of the work performed. Thus, we evaluated the influence of the accumulation of working days on the increase in body temperature of the scapular and lumbar regions of workers in a wood processing industry located in tropical forest regions through applying thermography as an ergonomic tool.

## MATERIALS & METHODS

The study was carried out with workers from a wood processing industry from a tropical forest located in the state of Mato Grosso in the central-west region of Brazil. The evaluated industry performs log splitting coming from native forest areas regulated by sustainable forest management into sawn wood, using specific techniques and machines for this purpose. The activities performed by the workers evaluated in this industry are described in Table 1.

A total of nine workers from the wood processing industry participated in the study, being firstly selected according to their acceptance to participate in the study, and then considering the exclusion criteria for participation (individuals away from work for more than 30 days, performing the function less than a year, using medication, diagnosed with

**Table 2** Average values of anthropometric variables of workers in the wood processing industry evaluated.

|  | Age (years) | Body mass (kg) | Height (m) | BMI (kg m$^2$)[a] |
|---|---|---|---|---|
| Average | 39 (21–58)[b] | 65.4 (60–76) | 1.73 (1.67–1.84) | 21.8 (20.7–24.8) |

**Notes.**
[a] Body mass index obtained by the formula: BMI = Weight/Height$^2$.
[b] Values in parentheses refer to the minimum and maximum values of each variable.

an injury or frequent consumers of alcoholic beverages). Table 2 shows the anthropometric characteristics of the evaluated workers.

## Thermographic assessment

The workers were previously instructed not to consume alcoholic beverages or caffeine, not to use any type of body moisturizer and not to practice vigorous physical exercises for a minimum period of 24 h prior to the assessment to obtain the thermographic images (*Moreira et al., 2017*). The collection of thermographic images was performed on two different days, in the morning (between 6:30 am and 8:30 am), as described below:

Day I—assessment carried out before the start of the workday on Monday: condition that considered the day on which the worker returns to work after a rest period during the weekend (>48 h).

Day II—evaluation carried out before the beginning of the workday on Friday: condition that considered the accumulation of days worked between Monday and Thursday, associated with a shorter rest time (corresponding to the end of the workday from Thursday to fair and start of the day on Friday, <12 h).

The evaluations were conducted on these two days with the aim of verifying whether the accumulation of days worked (day II) would provide a higher thermal temperature pattern in relation to the images obtained on Monday (day I), where the worker remained at rest from their duties for a greater time period. The posterior position of the workers' trunk was selected for the analysis; specifically, the body regions of the lumbar and scapular as defined based on the operation mode of the activities, as they are requested at all times and are more conducive to the development of injuries.

Figure 1 shows the lumbar and scapular regions selected for thermographic analysis, identified in the study as Body Regions of Interest (BRI).

The thermographic images were collected in a room equipped with a cooling air conditioner, where the participants remained with the upper part of their bodies bare (anterior and posterior regions of the hands, forearms and arms, abdominal, thoracic, cervical and lumbar regions) for an acclimatization period of 15 min with temperature conditions ranging from 22 °C to 23 °C, relative air humidity around 50% and air speed below 0.2 m s (*Marins et al., 2014b*; *Fernández-Cuevas et al., 2015*; *Costa et al., 2018*).

They were positioned away from any source of infrared radiation or airflow, the camera was turned on 30 min before the test to allow the sensor to stabilize, according to the manufacturer's guidelines, and the images were later captured perpendicularly to the body regions of interest (BRI).

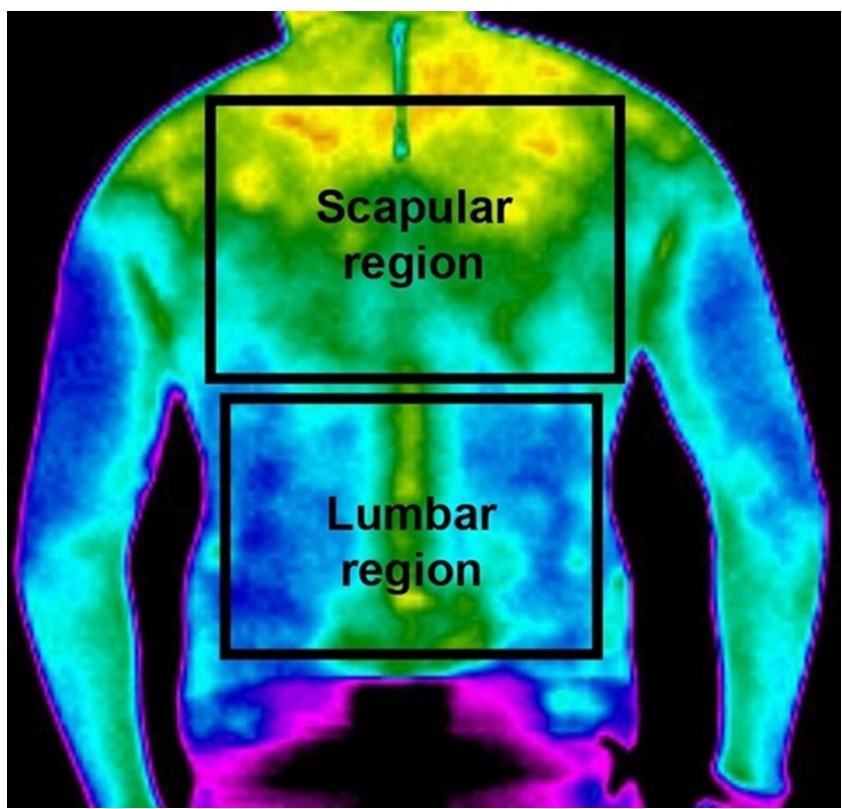

**Figure 1** Body regions of interest (BRI), lumbar and scapular, selected in the posterior position of each evaluated worker.

A Thermal Imager (FLUKE, model TI 400) was used to obtain the thermal images with a resolution of 320 × 240 pixels, an accuracy of ± 2 °C and an object temperature range from −25 °C to +105 °C. The thermographic images obtained were analyzed using the SmartView software program (version 3.1) through demarcation of the BRI (lumbar and scapular). The emissivity value adopted for human skin was 0.98 and the reflected temperature was set at 22 °C in the thermal camera.

Thus, the average temperature values of the selected BRI were considered for the quantitative analysis of the thermographic images, and the heat points visible in the thermogram were used for the qualitative (visual) analysis.

## Statistical procedures

Quantitative and qualitative evaluations were carried out in order to verify whether the accumulation of working days would provide a higher thermal temperature pattern in relation to the images obtained after a longer rest period.

The temperature data of the BRI were submitted to the Shapiro–Wilk normality test, and the paired Student's $t$-test was applied when the data were considered to have normal distribution to compare the average temperature of the BRI of the workers obtained on

**Table 3** Average and maximum skin temperature values of the Body Regions of Interest (BRI) of workers ($n = 09$) evaluated on day I and day II.

| BRI | | Temperature °C | |
|---|---|---|---|
| | | Day I | Day II |
| Scapular | Average | 32,70 | 32,40 |
| | Standard deviation | ±1,08 | ±0,90 |
| | Maximum | 34,2 | 33,9 |
| Lumbar | Average | 32,20 | 32,00 |
| | Standard deviation | ±0,70 | ±0,90 |
| | Maximum | 34,3 | 34,3 |

**Notes.**
The averages showed no significant statistical difference ( $p < 0.001$).

day I (image obtained after a longer rest period) versus day II (image obtained after the accumulation of days worked from Monday to Friday).

### Ethical principles

This study was approved by the Ethics Council of the Federal University of Espírito Santo (CAAE: 57864716.0.0000.5060/ approved on August 28, 2017), meeting the criteria established by Resolution No. 196/1996 of the Research Ethics Committee of the Ministry of Health from Brazil. Thus, we received the written consent of the participating workers by signing the Free and Informed Consent Form (*Federal Government of Brazil, 1996*).

## RESULTS

The analysis with the average and maximum values measured by the evaluation of thermographic images in the BRI (lumbar and scapular) of the workers is shown in Table 3.

The mean temperature values of the BRI (lumbar and scapular) of the workers evaluated on day I did not significantly differ in relation to day II, meaning there was no difference between the mean temperature values of the BRI evaluated due to a longer rest period (day I) and the accumulation of days worked (day II).

High temperature values regarding the qualitative analysis of thermographic images are noted, regardless of the day evaluated, located in the region corresponding to the spine, as shown in Fig. 2.

## DISCUSSION

This study aimed to evaluate the influence of the accumulation of working days on the increase in body temperature of the scapular and lumbar region of wood processing industry workers.

It was found that there was no significant difference between the average temperature values obtained in the thermographic evaluation from workers between day I and day II in the evaluated BRI. The results suggest rejecting the initially-raised hypothesis that the skin temperature of the evaluated body regions were higher at the end of the work week (Friday) in relation to the beginning of the work week (Monday). This was based on the idea that

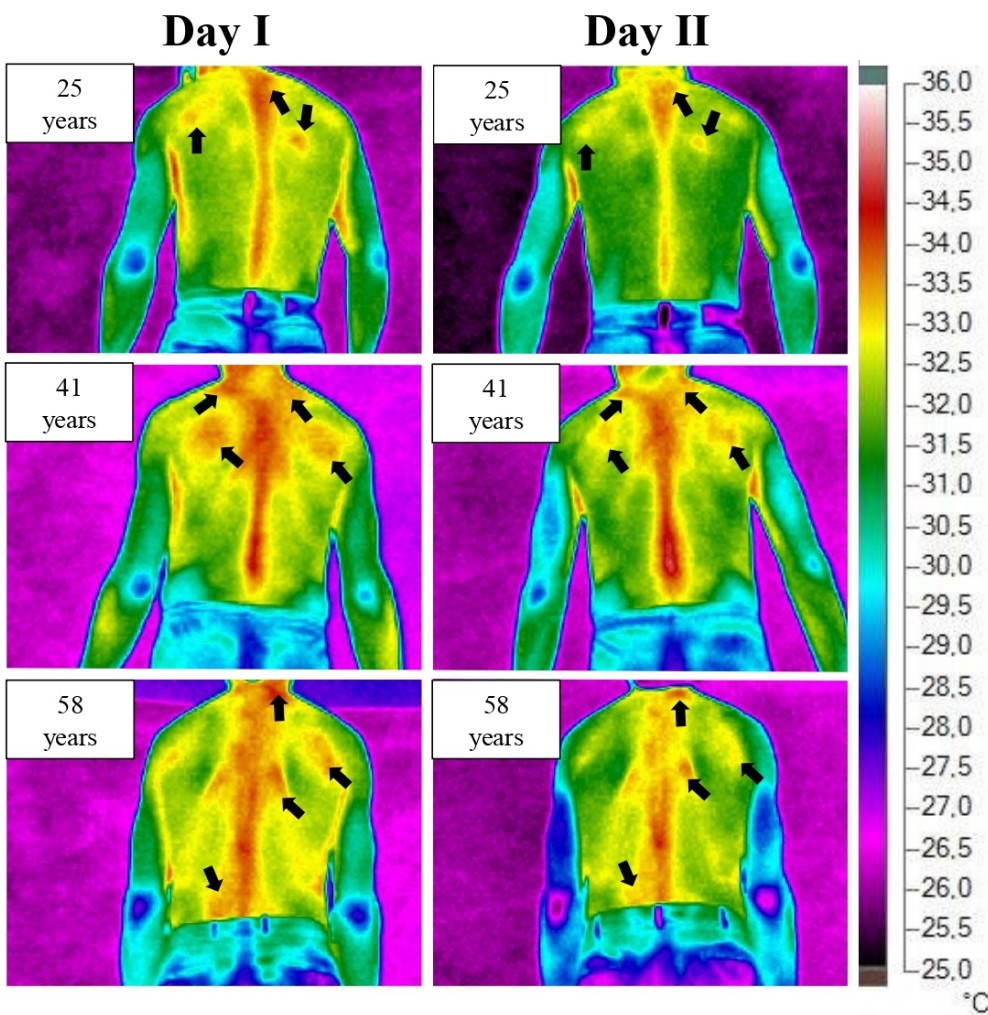

**Figure 2** **Comparison of thermal image of workers evaluated on day I (Monday, after a longer period of rest <48 h) and day II (rest of approximately 12 h associated with the accumulation of days worked).**

the worker could present a greater muscular overload resulting from the accumulation of days worked with a short rest interval (±12h). Therefore, the period equivalent to a weekend of rest (>48 h) may not be enough to evidence a possible muscle recovery, or that this thermal pattern may be the result of a musculoskeletal overload already predominant in the body regions evaluated in the workers.

An average temperature of 30.4 ± 0.7 °C in the lumbar region and 30.8 ± 0.7 °C in the scapular region was observed in a study carried out in Taiwan with 57 healthy individuals aged between 24 and 80 years (*Niu et al., 2001*). The respective values are between an average of 1.5 and 2 °C lower than those found in this study. In another study carried out with a population of young Brazilian men with a mean age of 21.3 ± 2.19 years, an average temperature of 31.56 °C ±1.0 was found in the lumbar region and 32.14 ± 0 .9 °C in the

upper region of the trunk, constituting values close to those reported in the workers in this study, but still lower (*Marins et al., 2014a*).

These studies reinforce the need to associate the use of thermography as an ergonomic analysis technique, as the data show evidence of a trend towards a higher temperature in the BRI (lumbar and scapular) of the evaluated workers. In addition, there is the fact that the evaluated workers perform extremely stressful activities. For example, the constant and repetitive lifting of heavy loads, such as sawn wood planks, long working hours lasting up to 10 h in some situations, in addition to the inadequate conditions of the environment and work station, which do not take into account the ergonomic principles recommended for the preservation of the worker's health.

Several factors interrelate which can cause musculoskeletal overload. Symptoms can be influenced by work techniques and organization, individual characteristics and psychosocial factors (*Cremasco et al., 2019*; *Tuček & Vaněček, 2020*). Repetitive movements and handling loads with high physical effort in situations in which workers are subjected to working hours that demand a biomechanical model of atypical postures can cause the development of musculoskeletal disorders, since this type of work rhythm exceeds the psychophysiological capacities of individuals (*Soranso et al., 2021*).

Such evidence is extremely widespread in the literature, which indicates that the probable causes of prevalence of musculoskeletal disorders in sawmill workers are verified due to repetitiveness at work, handling heavy loads and uncomfortable posture, which may be the possible factors causing the development of symptoms of musculoskeletal discomfort (*Adhikari & Sahu, 2018*; *Ajayeoba, 2019*; *Adetifa et al., 2020*; *Das, 2020*; *Jamaludin et al., 2022*).

Occupations with static postures and repetitive work are related to complaints of pain in the neck and shoulders, while problems in the lower back are often associated with heavy physical work with lifting loads (*Park et al., 2018*; *Tang et al., 2021*; *Sarker et al., 2021*).

The finding verified in the visual analysis of the images tends to be normal, since the lumbar and scapular are extremely vulnerable bone structures of the human organism (*Iida & Guimarães, 2016*), as they are used at all times by the human being, reducing their sustaining capacity over the years.

However, it can be seen that there was a prevalence of the thermal pattern among the images obtained from the workers on the two days evaluated. In other words, the places with the highest occurrence of maximum temperature in the evaluated BRI were the same on the first and second day of evaluation, with the occupation tending to be higher in workers with older ages.

Although the thermographic analysis performed in this study cannot provide definitive results, they helped to provide evidence for a more accurate diagnosis in the evaluated workers. Considering that this was an initial work aimed at the ergonomic evaluation of workers in the wood processing industry, and there is not much information on how to apply thermography in the evaluation of workers under these conditions, it is suggested to develop evaluation methods which enable analyzing the distribution of temperature within the body regions of interest for future works similar to this study.

 

In addition, it is noteworthy that even with no significant differences in the assessments used in this study, thermography proved to be a viable tool to detect thermal patterns, and it is possible to adopt its application as a monitoring tool in the preventive detection of possible health problems of workers. Its applicability in wood processing industry activities such as those evaluated in this study is highlighted by the great physical demand required of workers during the execution of the work, being a practical method and easy to apply.

## CONCLUSIONS

The accumulation of days worked associated with a shorter time for musculoskeletal recovery did not provide thermal patterns with significant differences in relation to the evaluation of workers after a longer rest period. The qualitative analysis showed a greater amplitude of occupation of thermal patterns with high temperature in the body region of interest of the workers in both situations evaluated.

The results generally indicate the need for an accurate examination in order to verify whether the development of wood processing operations contributes to musculoskeletal overload in workers. The study showed that the use of thermography associated with other assessment methods can contribute to improve the use of this technique in the ergonomic analysis of work through creating early and preventive assessment methodologies in monitoring the health and safety of workers who perform activities with great muscular demand, as is the case of those evaluated in this study.

### Funding
This study was financed by the Coordenação de Aperfeiçoamento de Pessoal de Nível Superior—Brasil (CAPES)—Finance Code 001. The funders had no role in study design, data collection and analysis, decision to publish, or preparation of the manuscript.

### Grant Disclosures
The following grant information was disclosed by the authors:
Coordenação de Aperfeiçoamento de Pessoal de Nível Superior—Brasil (CAPES)—Finance Code 001.

### Competing Interests
The authors declare there are no competing interests.

### Author Contributions
- Denise Ransolin Soranso conceived and designed the experiments, performed the experiments, analyzed the data, prepared figures and/or tables, authored or reviewed drafts of the article, and approved the final draft.
- Luciano José Minette conceived and designed the experiments, authored or reviewed drafts of the article, and approved the final draft.

- Marcio Marçal conceived and designed the experiments, authored or reviewed drafts of the article, and approved the final draft.
- João Carlos Bouzas Marins performed the experiments, authored or reviewed drafts of the article, and approved the final draft.
- Stanley Schettino performed the experiments, analyzed the data, authored or reviewed drafts of the article, and approved the final draft.
- Roldão Carlos A. Lima analyzed the data, prepared figures and/or tables, authored or reviewed drafts of the article, and approved the final draft.
- Michel Oliveira analyzed the data, prepared figures and/or tables, authored or reviewed drafts of the article, and approved the final draft.

## Human Ethics

The following information was supplied relating to ethical approvals (*i.e.*, approving body and any reference numbers):

This research was approved by the Ethics Council of the Federal University of Espírito Santo (CAAE: 57864716.0.0000.5060, approved on August 28, 2017).

## Field Study Permissions

The following information was supplied relating to field study approvals (*i.e.*, approving body and any reference numbers):

Field experiments were approved by the Ethics Council of the Federal University of Espírito Santo.

## Data Availability

The raw data are available in the Supplemental Files.

## Supplemental Information

Supplemental information for this article can be found online at http://dx.doi.org/10.7717/peerj.13973#supplemental-information.

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
