# Peer review of "Thermography in ergonomic assessment: a study of wood processing industry workers"

_PeerJ, doi:10.7717/peerj.13973_

## Round 0.1 · original submission · Major Revisions

Please, carefully address point-to-point all reviewers' issues.

Reviewer 1 ·

Basic reporting

no comment

Experimental design

no comment

Validity of the findings

no comment

Additional comments

General comments
This study evaluated the potentialities of thermal image assessment for the evaluation of body regions with musculoskeletal overload in wood workers. The study covers and interesting topic in the literature given the important practical perspectives for health of workers that might be derived from this study. Moreover, studies in exercise and movement sciences using thermal imaging are increasing continuously, as this topic might be considered a hot topic in the literature. Although the manuscript has merits and deserves attention, I recommend the Authors to substantially improve the overall quality of manuscript. Please see below some specific comments, but I remark here the most important ones, such as the coherence among the hypothesis and study aim, experimental protocol, statistical analysis, results presentation and main findings. A line should connect all these points. Moreover, language should be improved to ensure a clear understanding of the contents. For example, see the lines 82-85 that are difficult to follow (maybe also because of the wrong positions of commas). Another example is in line 93-94: this should be “the activities performed by the workers evaluated in this industry are described in Table 1”.

Specific comments
1. Line 60-61. “may be performed using thermography”. This requires citations to support this notion.
2. Line 60-65. I would recommend to include a description of the underlying physiological processes related to pain, that may be captured using thermography.
3. Line 82-85. The rationale behind the study aims is a crucial statement. Please rephrase it completely, as in this way is not clear.
4. Line 86-87. Aim statement. First, I do believe that the aim should be more specific. For example including times (1 h before and 1 h after?) and location (body regions) of assessment. What about the study hypothesis? Did authors expect something?
5. Table 2. Please include units of measurements of BMI.
6. Line 106. Did the Authors perform the thermographic assessment following the guidelines on thermal image in sport and exercise? Please refer to Moreira et al., 2017.

Moreira, D.G., Costello, J.T., Brito, C.J., Adamczyk, J.G., Ammer, K., Bach, A.J.E., Costa, C.M.A., Eglin, C., Fernandes, A.A., Fernández-Cuevas, I., Ferreira, J.J.A., Formenti, D., Fournet, D., Havenith, G., Howell, K., Jung, A., Kenny, G.P., Kolosovas-Machuca, E.S., Maley, M.J., Merla, A., Pascoe, D.D., Priego Quesada, J.I., Schwartz, R.G., Seixas, A.R.D., Selfe, J., Vainer, B.G., Sillero-Quintana, M., 2017. Thermographic imaging in sports and exercise medicine: A Delphi study and consensus statement on the measurement of human skin temperature. J. Therm. Biol. 69, 155–162. https://doi.org/10.1016/j.jtherbio.2017.07.006

7. Line 106. “Thermographic assessment” instead of “Obtaining thermal images”
8. Line 152-153. These should be included in the “thermographic assessement” paragraph. Also, please define RCI.
9. Line 168. “Quantitative”?
10. Table 3. (n=9). It is not clear at all the meanings of Aa, a and *. Please make it simple. “*Average of the same line followed by the same capital letter in bold does not differ (p<0.001).” Does not different with respect to what? * usually indicates a significant difference.
11. Figure 2 is clear and provides a straightforward visual impression of the temperature behaviour.
12. Line 184. I strongly suggest to begin the discussion section stating the main aim of the study and the main findings, that should reflect the statistical procedures adopted to answer to the experimental questions. Also, line 184-188 is a speculation. The main findings are that the temperatures of day I are not different with respect to the temperatures of day II. This comes from data and this should be clearly reported. Any sort of speculation and discussion should be furtherly reported.
13. Line 189-196. Ok for the absolute temperature, but maybe the potentialities of thermal imaging in the evaluation of posture and musculoskeletal disease should be found in relative temperature. For example in a pre-post comparison, or in a right-left comparison.

Reviewer 2 ·

Basic reporting

No comment

Experimental design

No comment

Validity of the findings

No comment

Additional comments

The paper reports about the capability of thermography to assess musculoskeletal overload in wood workers for ergonomic applications.
The paper is interesting and well written, however some concerns need to be addressed before publication:
1) Thermography is here used to assess muscular overload. Particularly, the measurements were performed in two different timings in order to assess the effect of the recovery on the muscular temperature. How did the Authors establish that the participants were affected by musculoskeletal overload? Is it supposed by the working condition of the workers?

2) Concerning the BRI and the metrics used to describe the temperature distribution within the BRIs, why did the Authors select only the lumbar and scapular regions? Did they investigate also the arms and the legs, for instance? Why only the average and maximum values of the temperature distribution were used for quantitative analysis? Since the qualitative analysis showed a different temperature distribution, it could be interesting to consider other statistical moments of the temperature distribution (e.g. kurostis, skewness, std) to quantitatively describe the changes associated to the work. For instance, the Authors could refer to the following paper which reports about the statistical metrics that could be used to describe the temperature distribution within the regions of interest.

• Perpetuini, D., Formenti, D., Cardone, D., Filippini, C., & Merla, A. (2021). Regions of interest selection and thermal imaging data analysis in sports and exercise science: a narrative review. Physiological Measurement.

3) In caption of table 3, it is not clear the meaning of Aa, a, *. Please better explain this aspect in the caption. Moreover, please report the t-stat, degree of freedom and p value of the comparison between the two measurements.

4) In the Discussion section, the Authors should stress the importance of the findings and the positive impacts for the wood workers, and generally, for ergonomics applications.

---

## Round 0.2 · Minor Revisions

Please, revise English language further.

Reviewer 1 ·

Basic reporting

no comment

Experimental design

no comment

Validity of the findings

no comment

Additional comments

The Authors have addressed all my comments. However, I would recommend to strongly improve the language before the publication.

Reviewer 2 ·

Basic reporting

No comment

Experimental design

No comment

Validity of the findings

No comment

Additional comments

The Authors addressed all my concerns and the manuscript is improved and, in my opinion, it is suitable for publication in the present form.

---

## Round 0.3 · accepted · Accept

Congratulations for the interesting work.